# Mechanical Alloying and Electrical Current-Assisted Sintering Adopted for In Situ Ti-TiB Metal Matrix Composite Processing

**DOI:** 10.3390/ma12040653

**Published:** 2019-02-21

**Authors:** Andrzej Miklaszewski, Mieczyslaw Jurczyk

**Affiliations:** Institute of Materials Science and Engineering, Poznan University of Technology, Jana Pawla II 24, 61-138 Poznan, Poland; andrzej.miklaszewski@put.poznan.pl

**Keywords:** mechanical alloying, nanoprecursor, electric pulse-assisted sintering, metal matrix composites

## Abstract

In this work, mechanical alloying and electrical current-assisted sintering was adopted for in situ metal matrix composite material processing. Applied at the initial powder stage, mechanical alloying enables a homogeneous distribution of the starting elements in the proposed precursor powder blends. The accompanying precursor preparation and the structurally confirmed size reduction allow obtainment of a nanoscale range for the objects to be sintered. The nano precursors aggregated in the micro-sized particle objects, subjected to electrical current-assisted sintering, characterize the metal matrix composite sinters with high uniformity, proper densification, and compaction response, as well as maintaining a nanoscale whose occurrence was confirmed by the appearance of the highly dispersed reinforcement phase in the examined Ti-TiB material example. The structural analysis of the sinters confirms the metal matrix composite arrangement and provides an additional quantitive data overview for the comparison of the processing conditions. The mechanical alloying examined in this work and the electrical current-assisted sintering approach allow in situ metal matrix composite structures to create their properties by careful control of the processing steps.

## 1. Introduction

For the growing expectations and challenges of modern engineering, Metal Matrix Composite (MMC) materials appear to remain a popular choice. The interest and significant attention accompanying the application of this material group always grows rapidly with the newly introduced general relations or the enhancement of the processing approach. The fields of present and prospective applications of the MMC materials are a result of their structure and their component distribution finds its source mainly in the high mechanical properties also available for high-temperature regimes. The properties of lightweight and corrosion-resistant MMC materials are good not only for military armors and automobile or aircraft engine parts [1,2,3] but also for biomaterials and hard tissue applications [4]. The investigations conducted in this work focused on the influence of the starting precursors (obtained by mechanical alloying) and current-assisted sintering provide the results that allow some new generalized conclusions for the MMC materials. The final composite material microstructure, through its above mentioned properties with their mutual relations, show an evident relation with the applied processing method.

The starting precursor compositions proposed in this work, based on titanium and boron powders, following the application of mechanical alloying (MA) and current-assisted sintering, allow the obtainment of a nearly fully compacted sinter composite structures, composed of a titanium matrix and a highly dispersed in situ nucleated titanium boride reinforcement phase. For the above type of MMC structures, other research considering different approaches investigated in literature and industrial practice [5,6,7,8,9] has not clearly indicated the relation between the processing method and the starting material, thus limiting the new possibilities of creation of the material properties.

For the proposed example of Ti-TiB metal matrix composite, the function of the phases remains strictly defined. Titanium, distinguished by its specific properties [10], shows a broad possible spectrum of applications. Considered as a matrix phase, it remains challenging during the processing due to its high reactivity with the interstitial elements such as O_2_, N_2_, C, or H_2_ contained in crucibles, the atmosphere or the material of the tools [11]. The selected reinforcement TiB phase appears as the best choice compared to other phases such as TiC, TiN, TiB_2_, SiC, B_4_C, Al_2_O_3_ or Si_3_N_4_ [12]. With its excellent mechanical properties, the thermal expansion coefficient close to that of the the matrix, and balanced thermochemical stability it is distinguished as the best in such a phase. For Ti + TiB composites, a significant improvement of properties could usually be obtained at the expense of the fracture toughness and ductility, especially at high volume fractions of the reinforcement phase [13], where interlocking may occur. The amount of the reinforcement phase should be precisely controlled knowing that its higher share in the composite structure may affect the phase morphology and size [14]. Research conducted by other investigators [15] shows that an average size of the TiB whiskers remains tightly connected with the Ti content in the initial mixtures. The same research also shows that the TiB whiskers in the composites synthesized from the mixtures containing low content of Ti powders have a high tendency to agglomerate. Besides the above, it is noteworthy that in general, the reinforcement phase could appear differently in a composite structure. For example, it could be added and mixed ex situ [16], transformed from another phase [17], or nucleated in situ [18]. The way that the reinforcement phase appears in the composite matrix classifies the applied approach directly by the required processing stages and the undertaken preparation steps or specific actions with its economic profitability and, finally, the material properties. The most important property for the MMC’s materials, however, remains the size of the precipitation phase, its amount, and the way of its distribution with the interface region sharpness as well as the overall features of the composite’s high density and the microstructure homogeneity. The processing approach that allows manipulating the above-mentioned features remains the best choice for recommendation to undertake a possibly quick industrial implementation path or to conduct breakthrough research in the field of interest.

The data analyzed and concluded to date have shown that much work has been done. However, sufficient attention has not yet been devoted to considering the starting material characteristics including the morphology and the size dimensions. The composition, purity, and morphology of the starting materials as essential densification factors regarding the process control, have not yet been analyzed in terms of the evaluation of the crystallite size range and their influence on the final structure and the obtained relation of properties. As will be confirmed in this work, the nanoscale could be preserved at the final product stage, after a properly defined material processing. Involving earlier data [19], the dimensions of the starting materials may influence the processing temperatures and the time periods as well as the resultant material properties. The obtained values of porosity and density for the compacts or the final sinters are also important and at the same time dependent on the selected processing path, different precursor material composition, or the starting dimension [18]. The density also has a distinct effect on the fracture toughness [12], which is why the presence of pores in any form should be avoided.

The ‘top-down’ mechanical milling approach applied for the required size transition of the starting materials by itself remains the source of additional effects. The pulverized or ripped reactor edge and vial material pollution as well as the aggregation effect that appears as a result of the powder (fragile or ductile) character remain commonly occurring problems in mechanically milled materials [20]. On the other hand, increased defect density, reduction of the particle size and a possible atomic penetration into the substitutional sites of the lattice with uniform dispersion of the starting components allow simplifying the diffusion processes also by their occurrence at the interfacial regions. For the nanoscale range structure with its inherent relation, the relevant material preparation and processing steps should be adhered to. High temperatures, long time periods of processing, or starting material contamination may often cause a grain growth-, phase-, or porosity- aggregation effect, as well as reduced properties in the final products. The electric current-assisted sintering method with its advantages [21,22] and proper processing control has the potential of overcoming the said limitations.

In this work, the innovative path of MMC material processing by mechanical alloying and current-assisted sintering was adopted. The analysis was carried out on the example mixtures of different nanoscale-sized precursor powder Ti-B compositions with homogenously distributed reactants to evaluate their influence on the final sintered composite properties. General relations were analyzed and conclusions drawn for the in situ MMC’s materials group based on the analyzed example. Uniform reinforcement phase dispersion was analyzed for the approach proposed in this work with its size in the microstructure and high densification and compaction response that contribute to the improvement of the composite properties.

## 2. Materials and Methods

Commercially pure titanium and boron powders (Alfa Aesar, Karlsruhe, Germany) with an average particle size of 45 and 80 µm respectively and a purity level of 99.5% were used for the precursor mixtures preparation. The starting powder morphology and the size were analyzed by the Scanning Electron Microscopy SEM (Vega-Tescan, Brno, Czech Republic) technique for control purposes. Three compositions of precursor blends proposed for the preparation of sinters with different weights of the starting elements were gathered and shown in Table 1 for higher data clarity.

The starting powders were weighed and mixed in the proposed proportions in the glove box (LabMaster 130, MBraun, Garching, Germany) under high purity argon atmosphere and poured into the hardened tool steel vial. Then, the blends were processed for 48 h, by mechanical alloying (MA) in a Spex Mixer Mill (SPEX SamplePrep, Metuchen, NJ, USA) with the ball-to-powder ratio (BPR) of 15:1. The obtained precursor compositions were structurally analyzed with Cu Kα radiation λ = 1.5406 Å (Panalytical Empyrean-Netherlands) in the range 30−80° to estimate the lattice parameters, the stress and the crystallite size after MA. The Willamson–Hall analysis was applied, where the size and strain induced broadening is deconvoluted by considering the peak width as a function of 2Θ [23] as:(1)$$B\cos\Theta =\frac{0.9\lambda }{D}+4\varepsilon \sin\Theta$$
where β is the full-width at half-maximum (FWHM) of the diffraction peak, Θ is the position of peak maximum, λ is the X-ray wavelength (λ = 0.15406 nm), D is crystallite size, and ε is the lattice strain. The linear plot course equation obtained for the proposed Uniform Deformation Model approach allows estimating the stress and the crystallite size of the analyzed precursor powder blend compositions.

In order to investigate the obtained precursor powder blends, the aggregated particle size, its morphology (according to GOST 25849 standard) and its distribution profiles, the SEM technique was repeatedly used.

For the sinters preparation, the obtained precursor compositions after synthesis were placed in a non-conductive ceramic die between two graphite punches with the initially applied force of 1 kN under vacuum conditions of 4 Pa according to the processing diagram shown in Figure 1 referred to as isolative die setup.

The Pulse Plasma Sintering (PPS) module equipment, designed and made by Elbit (Koszyce Wielkie, Poland), allows generating current pulses within a maximum stroke of 60 kA by a discharge of a battery unit capacitor (Elbit, Koszyce Wielkie, Poland), charged to a voltage of maximum 8 kV. The specific pulse duration and the frequency realize the sintering process that is automatically controlled by the measurement of the return temperature whose fluctuation is kept at ±35 °C. The voltage, the pulse frequency, the force, the temperature, and the real-time vacuum measurement remain under constant control during the process performed by a two-way software unit (Elbit, Koszyce Wielkie, Poland) with the latency of 0.1 s. Based on earlier research [18,19,24], the PPS process parameters were proposed for the preparation of sinters gathered and shown in Figure 2. The sample processing was realized using the ‘save/load’ mode function available in the upgraded software version (Prasa 1.26, Elbit, Koszyce Wielkie, Poland). For each proposed sintering temperature regime, 3 samples of the diameter of Ф10 mm were prepared.

A parameter setup including the die material, its geometry and size with the assumed punch distance and the material amount was loaded by the software operational module and was comparatively controlled for all the samples in terms of specific curves of the temperature relation. For further examination, the obtained samples after Pulse Plasma Sintering were ground and polished to a 1-μm finish.

The X-ray diffraction (XRD, Malvern Panalytical, Almelo, Netherlands) method was used for the sample structural characterization. The collected data were refined to estimate the structural parameters and variables such as the background, the profile coefficients, the lattice parameters and the linear absorption coefficients. The instrumental broadening effect for the collected data was eliminated by subtracting the full width at half-maximum (βo) of a standard (Si) sample from β of the respective Bragg peaks. The lattice parameter estimation as well as phase quantitative analysis were based on the Rietveld profile fitting method realized on the Maud software. The applied approach involved a simulation of the diffraction pattern based on the analyzed structural model for:Ti(α) (COD 9008517),TiB (COD 9008946)
The calculated pattern of the model structure was fitted to the observed one by minimizing the sum of the squares and after a refinement using the Marquardt least squares algorithm. For clarity, residual pattern indicators of the modeled data were revealed:Rwp—weighted pattern residual indicatorRexp—expected residual indicatorGOD—goodness of fit

The relative density of the obtained composite sinters was measured by the Archimedes drainage method in order to reveal the microstructure compactness. The porosity of the samples was estimated based on the observation of the optical microscopy images on the non-etched surfaces, realized in the bright and dark field mode on the Olympus (GX51, Olympus, Shinjuku, Tokio, Japan) unit, fitted with Plan Fluorite Objectives and a material solution plug-in dedicated software (Olympus, Shinjuku, Tokio, Japan). The microhardness measurements were carried out on the samples to determine the average hardness level by a Vickers (Innovatest Nexus 4302, INNOVATEST Europe BV, Maastricht, The Netherlands) tester within the 10 indents at an applied load of 300 g and loading time of 10 s. For microstructural observations, the samples were etched in the Kroll’s reagent and then characterized by the SEM technique.

## 3. Results

### 3.1. Precursor Powder Blend Stadium

The starting substrate elementary titanium and boron powders used for the blend composition preparation were characterized for the purposes of the processing control. The morphology and average particle size were determined from the SEM microphotographs shown in Figure 3 where the powder preparation stadium was presented schematically.

The average powder particle size, measured and confirmed with catalogue data, as shown in the attached size distribution profiles, is additionally characterized by an angular shape of the plate-like morphology with a higher size scatter for elementary boron and for a sharp-edged and flat surfaced angular-type titanium. After 48 h, the mechanically alloyed precursor powders, as shown on the SEM microphotographs, are characterized by spheroidal-type particle morphology and a growing angularity along with the boron amount. The additional particle diameter distribution profiles of the obtained precursors in the volumetric and count scales show an ongoing stadium of synthesis dependent on the boron amount that influences the powder average particle size and its scatter—from aggregation to fracture behavior. The profiles manifest a powder fragmentation stadium that occurs during processing and its relation with the boron amount during the synthesis also investigated and confirmed elsewhere [4,19]. The highly magnified SEM microphotographs show a powder particle aggregate structure after MA within smaller visible crystallite parts, also confirmed structurally.

The XRD spectra correlated and shown in Figure 4 show a strong relation between the structural response from the precursor material composition and its processing time.

The structural analysis of the obtained precursor compositions after mechanical alloying confirms the broadening of the growing peaks and their reduction due to the prolonged processing and the amount of boron. The starting boron amount influences the powder fragmentation characteristics during processing, depending on the strengthening mechanism of the reactants. After 48 h of synthesis, the precursors exhibited fragmentation, homogenization, and structure disintegration. The applied Willamson–Hall Uniform Deformation Model (UDM) approach allows an estimation (from the linear plots) of the crystallite size and the microstrain level related to the starting precursor composition after 48 h of MA. The functions in Figure 5 show a different course of the relation of the largest calculated crystallite size and the smallest microstrains for the 2% precursor sample to the smallest calculated crystallite size and the largest microstrains for the 10% precursor sample.

The nanoscale range confirmed in the obtained precursor compositions, dependent on the boron amount, indicates the same synthesis time, and advanced fragmentation stadium for the higher boron addition also confirmed in the synthesis time dependence in the previously conducted research [4,19]. The growing boron addition for the analyzed precursor powder examples, as the results confirm, increases the speed of precursors fragmentation, homogenization and structure disintegration reactions.

### 3.2. Sintered Composite Material Stadium

The precursor powder compositions, sintered in different temperature regimes by the PPS method, allowed obtaining metal matrix composite sinters with low porosity and high dispersion of the reinforcement phase in the nanoscale range.

The XRD spectra of the obtained sinters in Figure 6, confirm the composite structure in all the examined samples. The growing amount of boron in the starting precursor powder compositions (as reflected by the presented data for the obtained sinters by the change of ratio of the intensity of the peaks) increases the titanium boride precipitation phase with a simultaneous reduction of the peaks of the titanium matrix phase. The composite phase amount (PA) values estimated from the Rietveld analysis and presented in Table 2, also confirm the differences between the considered processing variants.

The crystallographic phase data for the titanium matrix also confirm the value of the c-axis parameter growing with the increase in the amount of boron in the precursor powder composition. The above relation remains directly connected with the structure boron intake, possible in accordance with the phase diagram which, for the maximum solubility, remains lower than 0.2 at. % for the hexagonal Ti(α) [25].

The microstructural analysis of sinters shown in Figure 7 confirms the obtained composite structure that consists of a titanium matrix and an in situ nucleated highly dispersed titanium boride reinforcement phase. The higher amount of boron in the starting precursor compositions, as seen in the microphotographs in Figure 7, increases the share of the reinforcement phase and leads to its visible coarsening in the obtained composite microstructure. The above growth at the same time corresponds to the reinforcement phase interlocking with the visible high dispersion in lower sintering temperature regimes. The grain size of the reinforcement phase, with the whisker-type morphology, falls in the nano to submicrometer scale range. Microstructurally compared different sinter treatment temperatures (900 and 1100 °C), also exhibit the precipitation phase clustering behavior in the higher proposed sintering regime for the obtained composite microstructure. The above supposition has its source in the interlocked area transformation. The locally occurring highly dispersed concentration regions of the precipitation phase remain an easy-to-transform microstructure area due to its energy-related favorability. The highest amount of boron in the starting precursor powders may also be recognized in the sinters microstructure by the locally occurring spatial boron diffusion centers described as regular in all-directions, spatial centers of the growth of the reinforcement phase.

The examined, low magnified images allow relating the starting precursor particle morphology, the size and composition with its processing temperature to the obtained porosity and microstructure of the sinters. The porosity depends on the amount of boron as well as on the PPS processing temperature and the occurring shapes and its distribution remains strongly related to the above. The higher boron amount in the starting precursors, revealed larger and less regular empty spaces in the obtained sinters (see Figure 7) thus influencing its compactness characteristics-see Figure 8.

The observed relation indicates a consistency/boundary regions for the sintered aggregated particles that could be modified by such diffusion moderators as temperature, composition, substrates’ size range, and lattice defects density. Higher-boron-amount precursors posses a higher potential for the reinforcement phase formation (composition imbalance + centers of the reaction between the substrates + diffusion kinetics), however, its growing amount blocks the movement of the atoms decreasing the speed of transformation and leading to its possible local concentration as well as microstructure locking. Higher-sintering-temperature regime decreases the appearance of the smaller voids in favor of the larger ones with their shape change to more regular during densification. The energy excess, as the microphotographs show, remains rapidly consumed by the matrix to remodel the microstructure. The amount of the reinforcement phase for the higher sintering temperature does not increase evidently, however, the characteristics of the compactness differs significantly (see Figure 8). The porosity change may thus become reduced by up to 50% by the processing temperature. The above-mentioned size, morphology, and composition of the starting precursor influence the compactness characteristics, the microstructure and the mutual relation between the final phase and its size range.

The densification and consolidation of the sinters proceeds under local plasma discharge conditions that occur at the particle grain boundaries as well as creates additional convection heating waves that spread in the whole material and remain mostly reflected in the ceramic die material protected by the boron nitride layer. The porosity in the material may change its share due to the starting precursor morphological relation, concentration of intrinsic structural defects, dimensions of the crystallite and diffusion mechanism energy relation dependent on the material composition as well as its processing conditions.

The performed microhardness test results analysis presented in Figure 9 for the obtained composite sinters, shows the preferred material response, dependent on the amount ratio of the precipitation phase. The above-mentioned high microhardness test results could be easily adjusted by controlling the starting precursor composition. The compared PPS temperature variant in the microhardness tests shows a slight difference among the obtained results, distinguishing, however, the high-temperature red side (1100 °C) followed by an increase in other fundamental properties of the sinters.

## 4. Discussion

As the present results confirm, the starting powder morphology, its average size and composition analyzed in the mechanical alloying processing stage, remain responsible for the final fundamental properties of the compacts or sinters. For example, the analyzed porosity share and its dispersion in the composite sinters microstructure obtained after the PPS procedure and visualized in Figure 7, justifies the attention focused on the morphological features of the particles to be sintered. The precursor powder compositions obtained after MA, characterized by different average sizes of the aggregates, scattering, as well as assumed morphology, remains influenced by the processing fragmentation mechanism. For the above, the growing boron amount in the starting powder blends composition, for which the cold welding and fracturing stages overlap, manifests itself by the crystallite size reduction with the simultaneous increasing in the microstrains. The differences in the observed porosity and relative density among the sinters distinguish the starting spheroidal morphology, higher processing temperature, and a lower boron addition to the starting precursor composition.

The obtained MMC sinters microstructure, shows relevant differences in the range of the analyzed variable amount of boron in the starting precursor compositions and the temperature processing setup. The amount of the precipitation phase, for example, corresponds not only to the variable amount of boron but also, as earlier data confirm [18], to its size promoted by its higher dispersion in the structure. According to the earlier presented data [18], micro-type precursor materials with the same compositions and processed by the same methods show nearly half the amount of the precipitation phase. The submicron and nano-sized whisker-type precipitation morphology that characterizes the microstructure of the sinters obtained in this work, for the growing amount of the reinforcement phase, distinguishes the occurrence of the interlocked centers. High dispersion of the precipitation phase, ensured by the crystallite size reduction and homogeneous elements distribution in the starting precursor compositions, finally leads to the obtainment of sinters of high uniformity. The presence of the interlocked area verified in the microstructure of all sinters, which most likely transform to the clustered area for the obtained processing temperature, remain responsible for the size growth of the reinforcement phase. Nonetheless, as other researchers have confirmed using the TEM and HRTEM methods, the in situ ceramic reinforcements, formed in the titanium or its alloys [26,27], generally exhibit a clean interface that promotes strong bonding between the phases allowing an obtainment of high properties of the composites. High microhardness test results could easily be adjusted by controlling the starting precursor composition as well as its size ratio as the previously presented results may confirm [18].

## 5. Conclusions

In this work, mechanical alloying and pulse plasma sintering methods were adopted for the processing of metal matrix composite materials. The applied approach allows for the observation and analyzing of the characteristics of the starting powder materials, their transformation, and their influence on the final properties of the composite materials. The following conclusions from the performed research can be drawn:the mechanical alloying approach allows a homogeneous distribution of the elements in the prepared precursor powder blends;the mechanical alloying approach allows obtaining the nanoscale range in the prepared precursor powder blends;the sinters obtained after PPS processing of the MA precursor powders are characterized by a composite microstructure with a highly dispersed reinforcement phase, whose size falls in the submicron to nanoscale range;the obtained in situ metal matrix composite sinters are characterized by low porosity, yet heavily dependent on the starting material composition characteristics and the proposed processing variants; andthe final material properties can be adjusted not only by the composition, the processing approach or the temperature dependence but also by the starting material morphology, its size ratio, and proper distribution of the components.

## Figures and Tables

**Figure 1 materials-12-00653-f001:**
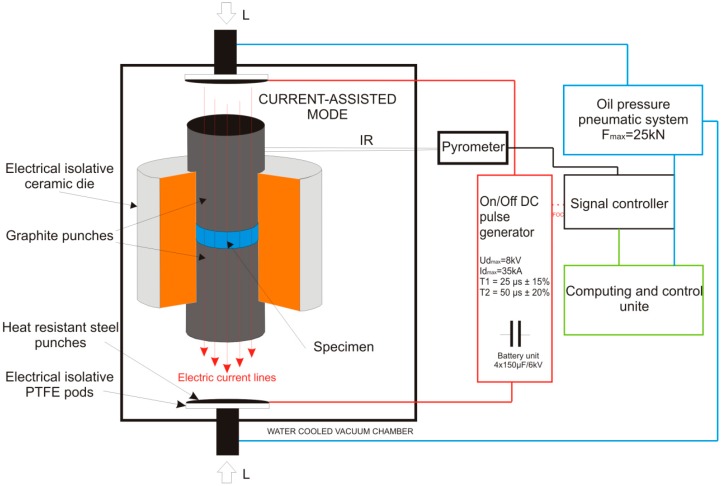
Schematic arrangement of the isolative die setup for a Pulse Plasma Sintering Press module.

**Figure 2 materials-12-00653-f002:**
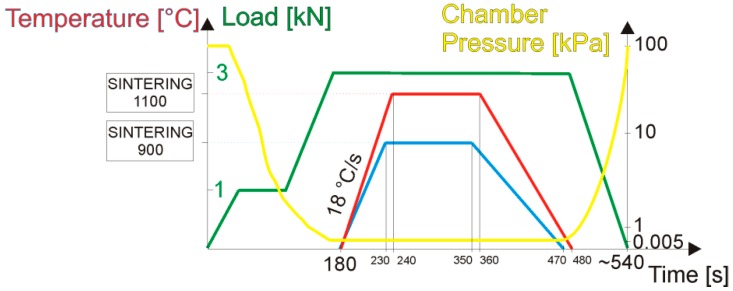
The Pulse Plasma Sintering processing parameter setup using the ‘save/ load’ operational function, presented in the form of a graph.

**Figure 3 materials-12-00653-f003:**
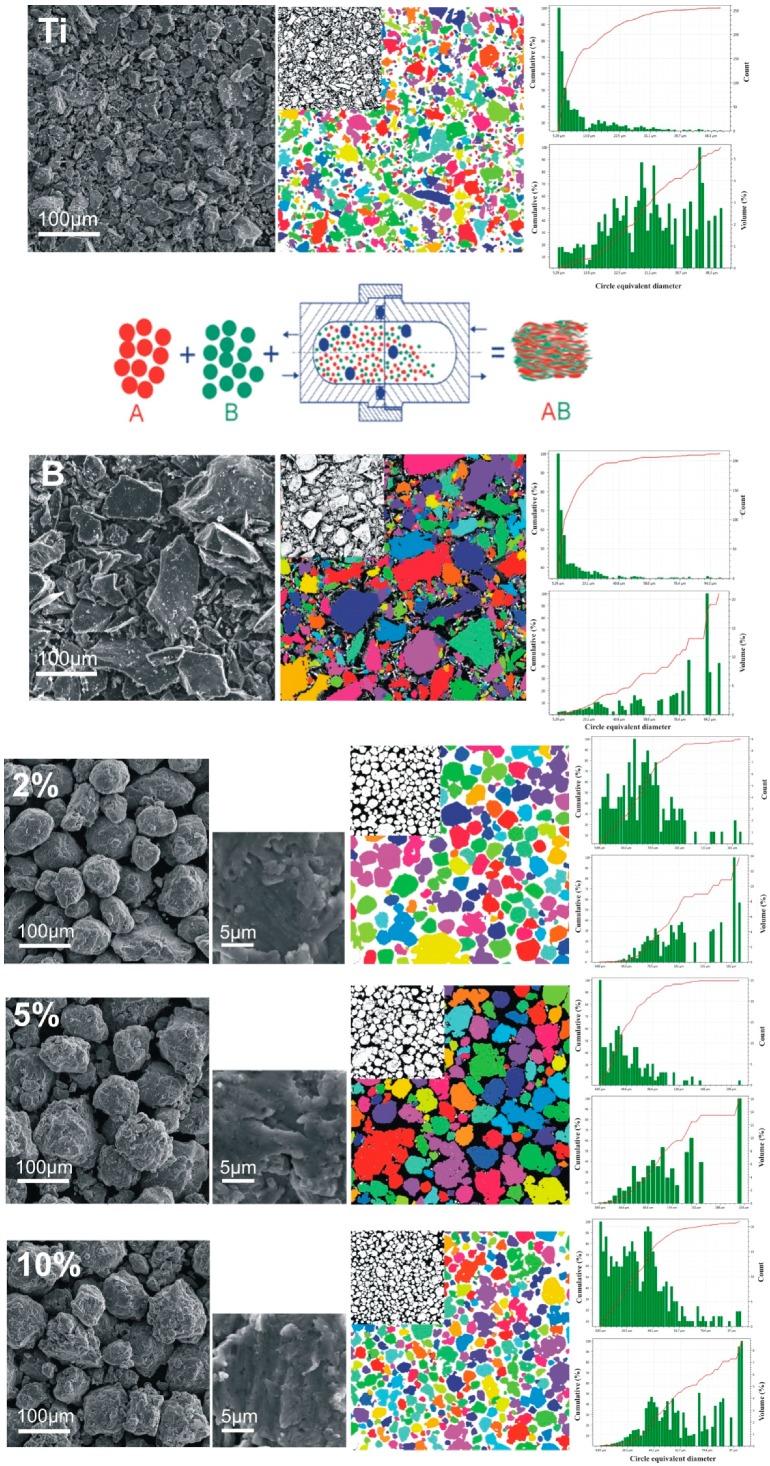
SEM microphotographs showing the powder blends processing stadium from the starting elementary titanium–Ti and boron–B powders to the 48 h mechanically alloyed precursor mixtures with different amounts of boron in the starting composition and their particle size distribution profiles in volumetric and count scales.

**Figure 4 materials-12-00653-f004:**
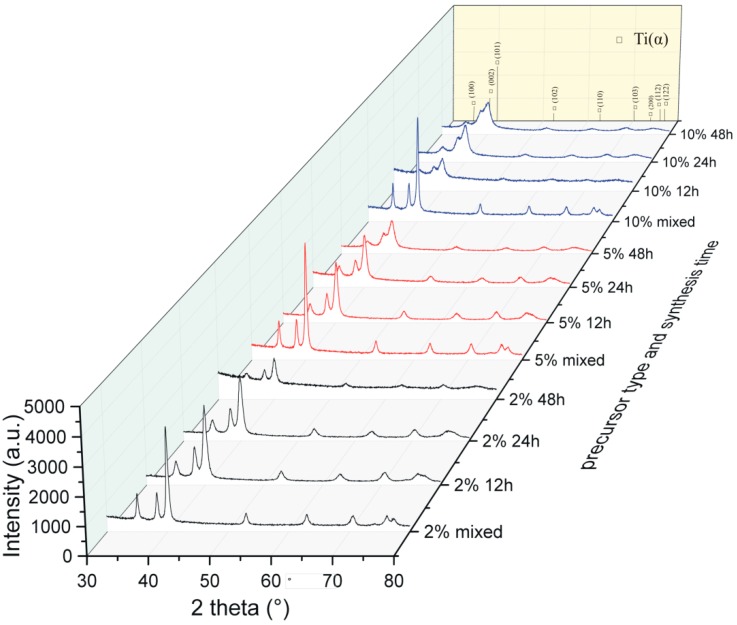
X-ray diffraction spectra of the precursors during mechanical alloying.

**Figure 5 materials-12-00653-f005:**
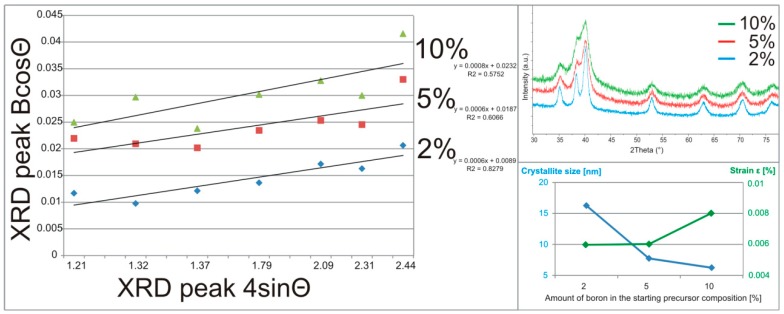
The crystallite size and the strain estimation based on the Willamson–Hall Uniform Deformation Model approach, realized on the XRD spectra of the obtained (after 48 h) mechanically alloyed precursor compositions with their linear plot course.

**Figure 6 materials-12-00653-f006:**
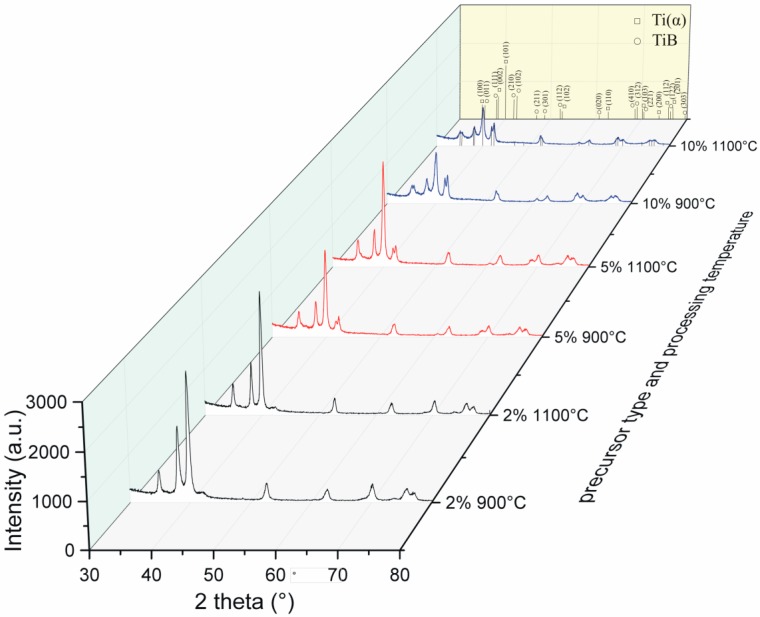
X-ray diffraction spectra of the obtained composite structures analyzed for different Pulse Plasma Sintering (PPS) temperature regimes and starting precursor compositions.

**Figure 7 materials-12-00653-f007:**
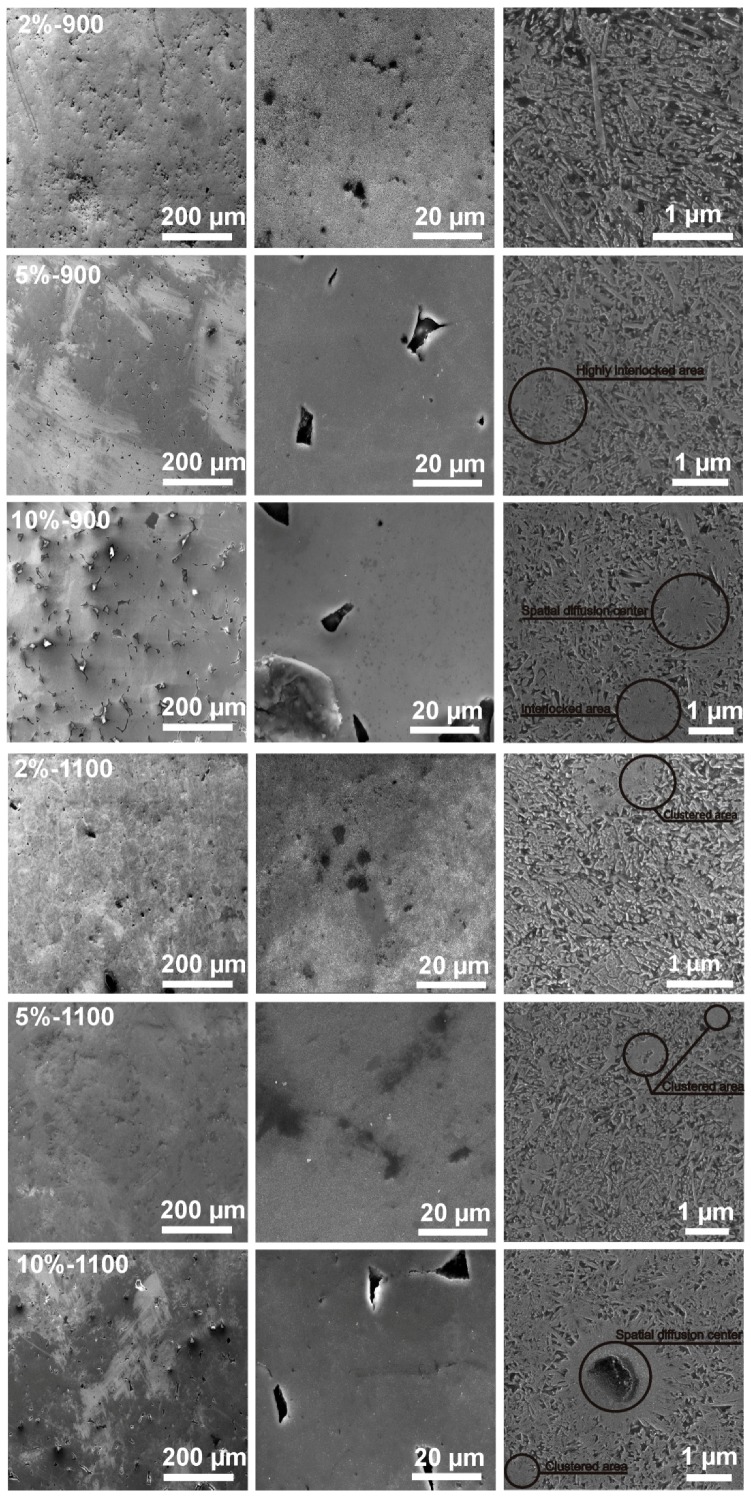
SEM microphotographs of the sinters obtained at different processing temperatures and from different starting precursor compositions (etched and non-etched surfaces).

**Figure 8 materials-12-00653-f008:**
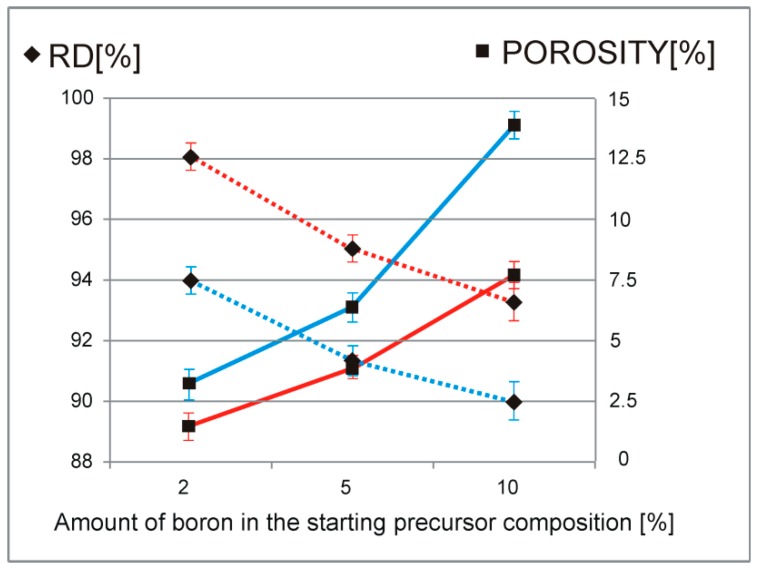
Relative density (RD–dotted lines) and porosity (full lines) analysis of the obtained composite sinters for different PPS temperature regimes (blue—900 °C, red—1100 °C) and ranges of the starting precursor compositions (2%, 5%, 10%).

**Figure 9 materials-12-00653-f009:**
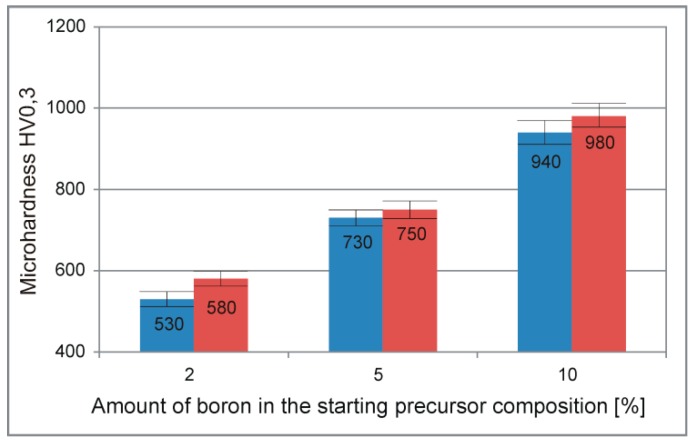
Microhardness analysis of the obtained composite sinters conducted for different PPS temperature regimes (blue—900 °C, red—1100 °C) and starting precursor compositions (2%, 5%, 10%).

**Table 1 materials-12-00653-t001:** Basic information about the starting element powders with the proposed precursor blends compositions and their indications.

Element	Average Particle Size [µm]	Purity Level [%]	Blend Composition 1 [wt.%]	Blend Composition 2 [wt.%]	Blend Composition 3 [wt.%]
Titanium	50	99.5	98	95	90
Boron	80	99.5	2	5	10
Symbol of the obtained after the MA process precursor composition			2%	5%	10%

**Table 2 materials-12-00653-t002:** Crystallographic data of the composite sinters obtained for different PPS temperature regimes (900, 1100 °C) and starting precursor compositions (2%, 5%, 10%), with the estimation of their phase amount and residual pattern indicators from the Rietveld approach analysis.

Treatment Temperature	Sample	Ti(α)			TiB				Rexp [%]	Rwp [%]	GOD
		a [Å]	c [Å]	PA [%]	a [Å]	b [Å]	c [Å]	PA [%]			
900	2%	2.9641(5)	4.7134(8)	89.96	4.5532(70)	6.0850(11)	3.0529(57)	10.04	8.67	19.67	2.27
	5%	2.9629(7)	4.7226(13)	68.82	4.5510(23)	6.0953(33)	3.0581(18)	31.18	7.04	19.73	2.80
	10%	2.9608(5)	4.7313(11)	42.63	4.5524(9)	6.1146(16)	3.0487(7)	57.36	9.06	14.10	1.56
1100	2%	2.9637(3)	4.7137(5)	90.65	4.5608(23)	6.1161(34)	3.0617(19)	9.35	9.22	15.55	1.69
	5%	2.9663(6)	4.7195(11)	70.58	4.5557(22)	6.1169(38)	3.0481(18)	29.42	7.29	19.18	2.63
	10%	2.9692(6)	4.7423(14)	40.59	4.5586(8)	6.1341(15)	3.0488(6)	59.41	9.57	16.66	1.74

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
