# Peer review of "Mechanical Alloying and Electrical Current-Assisted Sintering Adopted for In Situ Ti-TiB Metal Matrix Composite Processing"

_materials, 2019, doi:10.3390/ma12040653_

Round 1

Reviewer 1 Report

The article's many errors in English language usage make it difficult to understand.  In about a third of the sentences, I was able to understand the intended message.  In another third of the sentences, I found myself guessing about the intended message.  In the final third I was completely unable to understand what message was intended.  The experimental methods section was the easiest to understand; the other sections were more difficult to follow.

I suggest the authors work with a person who possesses strong English writing skills to edit and correct the existing document.  It is too rough at present to review or consider for publication.  From what I could understand from the narrative and the figures, it appears that the authors' study has scientific value, and I would be pleased to review the article if it is re-submitted to the journal in a more readable form.

Author Response

Dear Reviewer

Thank you for your comments concerning our manuscript. We have studied the comments carefully and have made correction (all correction marked in manuscript in RED ) .

Response for the reviewer comments

The article's many errors in English language usage make it difficult to understand.  In about a third of the sentences, I was able to understand the intended message.  In another third of the sentences, I found myself guessing about the intended message. In the final third I was completely unable to understand what message was intended. The experimental methods section was the easiest to understand; the other sections were more difficult to follow.

I suggest the authors work with a person who possesses strong English writing skills to edit and correct the existing document. It is too rough at present to review or consider for publication.  From what I could understand from the narrative and the figures, it appears that the authors' study has scientific value, and I would be pleased to review the article if it is re-submitted to the journal in a more readable form.

The author's one more time carefully revised the manuscript, the language correction was made by a native speaker, all of them with other changes in the text were marked in red.

We hope that our contribution in the present form could be considered for publication in Materials.

Sincerely yours Mieczysław Jurczyk

Reviewer 2 Report

Dear Authors,

I read carefully the manuscript entitled "Mechanical alloying and the electrical current assisted sintering adopt for the in situ Ti-TiB metal matrix composite processing". I find the research interesting and useful for all those studying this kind of materials.

Unfortunately, the major drawback of your manuscript is the English language. Besides the grammatical mistakes, you are using inappropriate terms (depictured/depicturing, synthesis time etc.) and expressions (Ti-TiB composite example). The title is long and has a mistake (you should write "...adopted for... instead of ".... adopt for").

You have to define all the acronyms where they are first used. Some of the acronyms are defined, but some are not (e.g. XRD, PPS). 

The type of the materials and the producer have to be removed from the first table. 

Pay attention to Figure 2: there is a mistake (pressure); do not use color in the boxes for the sintering temperature.

Figures 3 and 7: the images are too small; the reader cannot understand what the graphs are presenting; put fewer but clear images.

Figures 4 and 6: the graph is too small; we cannot read what you indicated for each peak.

Figure 5: you have to indicate the unit of measurement for each parameter; the second and the third graphs need to be magnified.

Figure 9: on the x-coordinate, if you have an amount, the unit of measurement is %; if you write 2B, 5B and 10 B are the precursors symbolizations; the values of the microhardness should be written above the columns.

You also have to pay attention at the figures and table captions.

I recommend you to rewrite the manuscript and send it again. At this moment, I do not believe that it can be accepted for publication.

Best regards.

Author Response

Dear Reviewer

Thank you for your comments concerning our manuscript. We have studied the comments carefully and have made correction (all correction marked in manuscript in RED )

Response for the reviewer comments

I read carefully the manuscript entitled "Mechanical alloying and the electrical current assisted sintering adopt for the in situ Ti-TiB metal matrix composite processing". I find the research interesting and useful for all those studying this kind of materials. Unfortunately, the major drawback of your manuscript is the English language. Besides the grammatical mistakes, you are using inappropriate terms (depictured/depicturing, synthesis time etc.) and expressions (Ti-TiB composite example). The title is long and has a mistake (you should write "...adopted for... instead of ".... adopt for").

The author's one more time carefully revised the manuscript, the language correction was made by a native speaker, all of them with other changes in the text were marked in red.

You have to define all the acronyms where they are first used. Some of the acronyms are defined, but some are not (e.g. XRD, PPS).

The authors defined the acronyms in the manuscript text

PPS – Pulse Plasma Sintering (p.4 l.147)

XRD - X-ray diffraction (p.5 l.167)

The type of the materials and the producer have to be removed from the first table.

The authors remove the type of materials and the producer from the Table1

Pay attention to Figure 2: there is a mistake (pressure); do not use color in the boxes for the sintering temperature.

The authors correct Figure 2 according to the reviewer suggestions

Figures 3 and 7: the images are too small; the reader cannot understand what the graphs are presenting; put fewer but clear images.

            The authors correct and enlarge Figure 3 and 7 according to the reviewer suggestions

Figures 4 and 6: the graph is too small; we cannot read what you indicated for each peak.

            The authors correct and enlarge Figure 4 and 6 according to the reviewer suggestions

Figure 5: you have to indicate the unit of measurement for each parameter; the second and the third graphs need to be magnified.

            The authors correct and enlarge Figure 5 according to the reviewer suggestions

Figure 9: on the x-coordinate, if you have an amount, the unit of measurement is %; if you write 2B, 5B and 10 B are the precursors symbolizations; the values of the microhardness should be written above the columns.

The authors correct Figure 8 and 9 as also graphical abstract according to the reviewer suggestions

You also have to pay attention at the figures and table captions.

            The authors correct the figures and table captions according to reviewer suggestion

I recommend you to rewrite the manuscript and send it again. At this moment, I do not believe that it can be accepted for publication.

We hope that our contribution in the present form could be considered for publication in Materials.

Sincerely yours Mieczysław Jurczyk

Reviewer 3 Report

The paper in the present form requires some modifications including the language. Some of the lacunae and possible modifications are given below:

·         There are several grammatical errors and unclear sentences throughout the manuscript including the abstract, the language of the whole manuscript needs to be carefully reviewed and improved. Authors may seek the help of a writing center or a native English-speaking person.

·         The introduction section is long. However, it does not discuss the literature in a systematic and smooth manner and it does not provide clear reasoning of this work. The structure and flow of the introduction section need to be changed to address these issues.

·         Proper numbering is needed for the sections (titles) and subsections (subtitles) throughout the manuscript.

·         For the PPS acronym, the whole term “Pulse Plasma Sintering” needs to be added the first time the PPS acronym is being used. Please make sure that this is the case for all other used acronyms throughout the manuscript.

·         Figure 3 contains a lot of information and the caption is very confusing. Please rewrite the caption and modify the figure as needed to make it easy for the reader to understand the content of the figure. Also, it could be helpful to split Figure 3 into two figures for a better representation of the results.

·         Similarly, for Figure 7, it is very hard to understand the presented results. The caption should be able to stand by itself and explain what is seen in the figure without the need to read the text.

Author Response

Dear Reviewer

Thank you for your comments concerning our manuscript. We have studied the comments carefully and have made correction (all correction marked in manuscript in RED )

Response for the reviewer comments

The paper in the present form requires some modifications including the language. Some of the lacunae and possible modifications are given below:

There are several grammatical errors and unclear sentences throughout the manuscript including the abstract, the language of the whole manuscript needs to be carefully reviewed and improved. Authors may seek the help of a writing center or a native English-speaking person.

The introduction section is long. However, it does not discuss the literature in a systematic and smooth manner and it does not provide clear reasoning of this work. The structure and flow of the introduction section need to be changed to address these issues.

The author's one more time carefully revised the manuscript, the language correction was made by a native speaker, all of them with other changes in the text were marked in red.

Proper numbering is needed for the sections (titles) and subsections (subtitles) throughout the manuscript.

            The authors introduce the numbering of the sections according to the reviewer suggestions

For the PPS acronym, the whole term “Pulse Plasma Sintering” needs to be added the first time the PPS acronym is being used. Please make sure that this is the case for all other used acronyms throughout the manuscript.

The authors defined the acronyms in the manuscript text

PPS – Pulse Plasma Sintering (p.4 l.147)

XRD - X-ray diffraction (p.5 l.167)

Figure 3 contains a lot of information and the caption is very confusing. Please rewrite the caption and modify the figure as needed to make it easy for the reader to understand the content of the figure. Also, it could be helpful to split Figure 3 into two figures for a better representation of the results.

            The authors correct and enlarge Figure 3 according to the reviewer suggestions      

Similarly, for Figure 7, it is very hard to understand the presented results. The caption should be able to stand by itself and explain what is seen in the figure without the need to read the text.

The authors correct and enlarge Figure 3 and 7 according to the reviewer suggestions

We hope that our contribution in the present form could be considered for publication in Materials.

Sincerely yours Mieczysław Jurczyk

Round 2

Reviewer 2 Report

Dear Authors,

I read carefully the new version of the manuscript entitled "Mechanical alloying and the electrical current assisted sintering adopt for the in situ Ti-TiB metal matrix composite processing".

I saw that you have taken into account the observations and suggestions I made. You definitely have improved the quality of the manuscript. You have also improved the English language.

I believe that now the manuscript may be considered for publication in "Materials".

Best regards.